# One-Pass Distribution Sketch for Measuring Data Heterogeneity in Federated Learning

**Zichang Liu**[*]
Rice University
zichangliu@rice.edu

**Zhaozhuo Xu**[*]
Stevens Institute of Technology
zxu79@stevens.edu

**Benjamin Coleman**
Rice University[†]
benjamin.ray.coleman@gmail.com

**Anshumali Shrivastava**
Rice University & ThirdAI Corp.
anshumali@rice.edu

## Abstract

Federated learning (FL) is a machine learning paradigm where multiple client devices train models collaboratively without data exchange. Data heterogeneity problem is naturally inherited in FL since data in different clients follow diverse distributions. To mitigate the negative influence of data heterogeneity, we need to start by measuring it across clients. However, the efficient measurement between distributions is a challenging problem, especially in high dimensionality. In this paper, we propose a one-pass distribution sketch to represent the client data distribution. Our sketching algorithm only requires a single pass of the client data, which is efficient in terms of time and memory. Moreover, we show in both theory and practice that the distance between two distribution sketches represents the divergence between their corresponding distributions. Furthermore, we demonstrate with extensive experiments that our distribution sketch improves the client selection in the FL training. We also showcase that our distribution sketch is an efficient solution to the cold start problem in FL for new clients with unlabeled data.

## 1 Introduction

Federated learning (FL) has recently become the go-to solution for challenging on-device applications such as typing prediction [1], machine translation [2, 3] and fraud detection [4, 5]. Initially proposed by McMahan et. al. [6], the FL setting defines a learning problem where multiple clients collaborate to construct a global model from local client data. Unlike in centralized learning, where clients can freely exchange data, FL requires that private data never leave the client device. The FL framework is uniquely positioned to comply with recent data privacy legislation [7, 8, 9], and it has correspondingly seen a surge of interest in machine learning (ML) research and deployment [10, 11, 12].

**Data Heterogeneity in FL.** Clients' local datasets may be drawn from significantly different statistical distributions [6, 13, 14, 12, 15, 16, 17, 18, 19, 20, 21, 22]. For instance, in the typing prediction task, each client is a cellphone user. Differences in user interest, language, and geography will produce extremely diverse word distributions. Data heterogeneity is a severe problem for FL because the models trained on each client may have many different behaviors. It is challenging to aggregate such different client models into a coherent and performant global model.

---

[*]Equal contribution. The order of authors is determined by flipping a coin.
[†]Now with Google DeepMind.
 Code is available at https://github.com/lzcemma/RACE_Distance

**Benefits of Measuring Data Heterogeneity.** We currently lack good methods to measure data heterogeneity – but we expect that such methods could substantially improve the efficiency and quality of FL [23, 24, 25]. We will use the term *global distribution* to denote the combination of client data distributions. If a client's data distribution is close to the global distribution, we can obtain a better global model by focusing training resources on this client (improving efficiency). Data heterogeneity measures can also help implement an essential personalization procedure of FL [26, 27, 28]. In personalized FL, each client increases local predictive performance by finetuning the global model on their own data. This results in a personalized model for each client. Now, suppose that a new client arrives with unlabeled data. We can search for a client who has data that is similar to the new client and use its personalized model. To summarize, data heterogeneity measurement allows us to quantify the differences between clients in FL to improve FL algorithms and systems.

**Challenges in Measuring Data Heterogeneity in FL.** The data heterogeneity measurement can be formulated as a statistical problem where we wish to quantify the degree to which probability distributions differ. However, this divergence estimation can be difficult in high-dimensional space. Existing approaches, including earth mover distance [29, 30] or Fréchet distance [31, 32] are computationally expensive. Moreover, many classical algorithms fail for complex high-dimensional input distributions; learning is much more effective [32]. However, learning introduces extra computational overheads. Since the clients may be mobile devices with limited hardware resources in the FL setting, expensive computation will introduce a significant efficiency issue in practical deployment. Further, to keep with the privacy advantage of FL, data heterogeneity measurements must to private.

## 1.1 Our Proposal: One-Pass Distribution Sketch with Hashing

In this paper, we propose an efficient approach to represent client data distributions in FL. Motivated by the recent success of locality sensitive hashing in kernel density estimation, we develop a one-pass distribution sketch for measuring data heterogeneity on FL clients. We make the following specific contributions.

1. We propose an effective and efficient distribution sketching method that requires only one pass over the client data. We represent the data distribution with a matrix of counts, and we introduce a data heterogeneity measure which is the Euclidean distance between clients' sketch matrices.
2. We develop an efficient sketch-based sampling strategy for federated optimization. Given sketches for each FL client, we find a global distribution sketch by merging all client sketches. We propose a form of client importance sampling where, in each training round, we preferentially select clients that have a low distance to the global sketch. We observe a significant acceleration in FL, with faster convergence and better generalization than uniform sampling.
3. We develop an efficient cold start approach for personalized FL where a new client (with unlabeled data) selects the most appropriate model from the personalized finetuned models of other clients. To do this, we compare the distribution sketch of the new client with each existing client. We perform a nearest neighbor search over clients under the sketch distance to identify the client that has similar data distribution. Our cold-start approach provides a simple yet effective way to find a suitable model for new clients.

## 2 Preliminary

**Locality Sensitive Hashing (LSH):** LSH functions serve as a key ingredient in our distribution sketch. LSH is a randomized function family [33, 34, 35, 36, 37]. Each function in this family maps an input vector into a hash value, usually binary code or integer. If two vectors are close to each other, with a high probability they will share the same hash value. Formally, we define the LSH function as:

**Definition 2.1** $((D, cD, p_1, p_2)$**-sensitive hash family**)**.** *We define a function family $\mathcal{H}$ to be $(D, cD, p_1, p_2)$-sensitive with respect to distance function $d : \mathbb{R}^d \times \mathbb{R}^d \to \mathbb{R}$ if for any two vector $x, y \in \mathbb{R}^d$, any $h \in \mathcal{H}$ chosen uniformly at random satisfies:*

- *If $d(x, y) \leq D$ then $Pr[h(x) = h(y)] \geq p_1$*
- *If $d(x, y) \geq cD$ then $Pr[h(x) = h(y)] \leq p_2$*

Here $D \in \mathbb{R}$ and $c > 1$. We denote $Pr[h(x) = h(y)]$ as the collision probability of vector $x$ and $y$. Here the term "collision" means $x$ and $y$ have the same hash value. In practice, we will use LSH functions where the collision probability between $x$ and $y$ is a monotonically increasing function with respect to a distance measure $d : \mathbb{R}^d \times \mathbb{R}^d \to \mathbb{R}$. Formally, we denote this relationship as:

$$Pr[h(x) = h(y)] \propto f(d(x, y)). \tag{1}$$

We introduce the realization of the LSH function in Section A of the Appendix.

**Repeated Array of Count Estimators (RACE):** RACE is an efficient Kernel Density Estimator (KDE) with applications in outlier detection [38, 39], graph compression [39], genetic profiling [40] and continue learning [41]. RACE builds a two-dimensional matrix $A \in \mathbb{R}^{R \times B}$ using $R$ LSH functions (see Definition 2.1), each LSH function $h_i$ maps a vector in $\mathbb{R}^d$ to an integer in $[B]$. Given a dataset $\mathcal{D} \subset \mathbb{R}^d$, for every $x \in \mathcal{D}$, RACE maps it to $R$ different hash values, denoted as $\{h_i(x)|i \in [R]\}$. Then we increment entry $A_{i,h_i(x)}$ with 1 for every $i \in [R]$. Following this manner, RACE maintains a matrix where each column can be regarded as a histogram of the dataset $\mathcal{D}$. As a result, we can use matrix $A$ for KDE and its downstream tasks.

# 3 One-Pass Distribution Sketch

In this section, we introduce the one-pass distribution sketch for measuring data heterogeneity in FL.

## 3.1 One-Pass Sketch Construction

**Intuition:** We observe that the RACE matrix can be used as a representation for a data distribution. Given a dataset $\mathcal{D} \subset \mathbb{R}^d$, if we build a RACE matrix $S$ by inserting every element in $\mathcal{D}$, we build a histogram of data samples with hashing-based projection. Next, if we normalize the counts in the RACE matrix $S$ with the dataset size $|\mathcal{D}|$, we will get a representation of $\mathcal{D}$.

Following this intuition, we introduce our distribution sketch algorithm in Algorithm 1. Our distribution sketch $S$ is a two-dimensional matrix with $R$ rows and $B$ columns. Given a dataset $\mathcal{D}$, we first initialize a zero matrix $S \in \mathbb{R}^{R \times B}$ and $R$ LSH functions $\{h_1, \cdots, h_R\}$. Here, each hash function $h_i$ generates an integer hash value within $[B]$. We suggest $R$ to be $O(\sqrt{|\mathcal{D}|})$. The choice of $B$ depends on the LSH functions we use. Next, for every $x \in \mathcal{D}$, we compute $R$ hash values $\{h_1(x), \cdots, h_R(x)\}$. Finally, we increment $S_{i,h_i(x)}$ for every $i \in [R]$. In practice, for vision tasks with dense feature vectors, we use SRP hash (See Definition A.1) as our LSH function. For language modeling tasks, we tokenize every sentence in the dataset into a bag of

---

**Algorithm 1** One-Pass Distribution Sketch

**Input:** Dataset $\mathcal{D} \subset \mathbb{R}^d$, LSH function family $\mathcal{H}$, Hash range $B$, Rows $R$, Random Seed $s$
**Output:** Sketch $S \in \mathbb{R}^{R \times B}$
**Initialize:** $S \leftarrow 0^{R \times B}$
Generated $R$ independent LSH functions $h_1, \ldots, h_R$ from $\mathcal{H}$ with range $B$ and random seed $s$.
**for** $x \in \mathcal{D}$ **do**
    **for** $i = 1 \to R$ **do**
        $S_{i,h_i(x)}+ = 1$
    **end for**
**end for**
$S = S/|\mathcal{D}|$
**return** $S$

---

words or characters [42]. Next, a sentence can be formed as a binary vector with a nonzero index representing a token. In this case, we can use MinHash (see Definition A.2) as an LSH function.

**Complexity Analysis:** Given a dataset with $n$ vectors, the running time of Algorithm 1 is $O(nRd + RB)$. Since the number of LSH functions $R$ and hash range $B$ are usually smaller than the dataset size $n$, Algorithm 1 is a linear passing over the dataset. Moreover, the space complexity of Algorithm 1 is $O(RB)$, which is a much smaller than the memory cost of storing the dataset.

## 3.2 Theoretical Analysis

In this section, we describe how to measure the data heterogeneity using the one-pass distribution sketch in Algorithm 1. We will estimate the total variation distance (TVD), which roughly measures the largest pointwise difference between two distributions over their support.

**Definition 3.1.** *Given two probability measures $P$ and $Q$ on a space $(\Omega, \mathcal{F})$,*

$$\text{TVD}(P, Q) = \sup_{A \in \mathcal{F}} |P(A) - Q(A)| \tag{2}$$

*If P and Q have density functions p and q, this expression is equivalent to the following $\ell_1$ norm.*

$$\text{TVD}(p, q) = \frac{1}{2} \int_\Omega |p - q| dv \tag{3}$$

Our goal is to describe a method for consistent estimation of the TVD that is efficient to implement in the distributed and private FL setting. At a high level, our strategy is as follows.

1. Given a dataset $D$ drawn from a distribution with probability density function $p(x)$, construct a sketch-based consistent estimator $S_D(x)$ for the kernel density estimate (KDE) over $D$.
2. Argue that $S_D$ is a consistent estimator of $p$, because the KDE is a consistent estimator of $p$ and $S_D$ is a consistent estimator of the KDE.
3. Use $S_D$ as a plug-in estimator for $p$ in the TVD integral (Equation 3).

**Assumptions:** We consider probability measures on $\mathbb{R}^d$ that have a distribution function. We assume that the probability densities $p$ and $q$ are continuous almost everywhere, so that the KDE $\rightarrow p$ in step 2 [43]. We also assume that the support of $P$ and $Q$ is a bounded subset of $\mathbb{R}^d$, so that our kernel estimator has the properties described in Section 3.2.1. Finally, our results are for convergence in probability rather than the (stronger) almost sure uniform convergence.

**Notation:** In a slight abuse of notation, we will use $O(\cdot)$ to denote both stochastic bounds and the more typical (deterministic) asymptotic bounds. More concretely, $X_N = O(f(N))$ means that for any $\epsilon > 0$, we can find constants $c$ and $N_0$ such that for all $N > N_0$, $\Pr[\|X\| \geq cf(N)] \leq \epsilon$. In the deterministic case, this still holds with $\epsilon = 0$. We will assume two probability measures $P, Q$ having densities $p, q$ and two datasets $\mathcal{D}_p \sim p$ and $\mathcal{D}_q \sim q$. We denote the sketch of $\mathcal{D}_p$ with $S_p$ and $\mathcal{D}_q$ with $S_q$. We write the distribution sketch $S$ generated by Algorithm 1 as a functional $S(x)$. We explicitly define this functional in Algorithm 4, Section D.

### 3.2.1 Consistent Estimation with Sketches

This section shows that our sketches provide a consistent estimator of the data-generating distribution. We introduce the distribution sketch's query algorithm in Algorithm 4 (See Section D in the appendix). For a sketch constructed via Algorithm 1, given a query $q \in \mathbb{R}^d$, we use the same LSH functions in construction sketch and get $R$ hash values. The $i$th hash value corresponds to a location in the $i$th row. Next, we take the median over the average counts in the hashed locations. We show that the median-of-means count $S(q)$ converges in probability to the kernel density of the LSH kernel. We begin with the definition of an LSH kernel.

**Definition 3.2** (LSH Kernel). *We define a kernel $k(x, y)$ to be an LSH kernel if for every $x, y$, $k(x, y)$ represents the LSH collision probability (See Eq. (1)) of $x$ and $y$, which is monotonically decreasing with distance $d(x, y) \geq 0$. An LSH kernel $k(x, y)$ is a positive semi-definite kernel.*

Next, we have the following statement.

**Theorem 3.3** (Informal version of Theorem D.1). *Let $P(x)$ denote a probability density function. Let $\mathcal{D} \underset{\text{iid}}{\sim} P(x)$ denote a dataset. Let $k(x, y)$ be an LSH kernel (see Definition 3.2). Let $S$ define the function implemented by Algorithm 4. We show that*

$$S(x) \underset{\text{i.p.}}{\rightarrow} \frac{1}{N} \sum_{x_i \in \mathcal{D}} k(x_i, q)$$

*with convergence rate $O(\sqrt{\log R / R})$.*

Theorem 3.3 shows the consistent estimation of the KDE as $R \rightarrow \infty$, which is itself a consistent estimator of the distribution under some assumptions about $k(x, y)$ and $P$. The convergence of the KDE to the distribution as $N \rightarrow \infty$ is a well-studied problem under various conditions [44]. However, for the weak form of convergence which we consider here, it is sufficient for the kernel to integrate to 1, for the kernel bandwidth to decay with $N$, and for the bandwidth to be bounded above by $N^{1/d}$ [45].

We argue that these conditions hold for the LSH kernels described by [46]. The bandwidth requirements are easy to satisfy because we can explicitly set the LSH kernel bandwidth by adjusting the

hash function parameters - for example, by setting the number of bits in an SRP hash. The other requirement can be tricky to satisfy for LSH kernels, such as the $p$-stable LSH probability, which is not absolutely integrable over $\mathbb{R}^d$. For this reason, some LSH kernels cannot be normalized to have a unit integral. However, this is not a problem because we assume that $P$ and $Q$ have bounded support in $\mathbb{R}^d$. Note that this type of input domain truncation is a standard component of LSH design and analysis [47]. Under these conditions, Theorem 3.3 implies consistent estimation of $P(x)$ by $S(x)$ provided that $R$ is allowed to weakly grow with $N$ (i.e. $R = \omega(1)$).

**Private Sketches:** When the sketch is required to preserve $\epsilon$-differential privacy, we must add independent Laplace noise to the sketch so that each counter is private [48]. This constrains our choice of $R$ because the noise is distributed $Z \overset{\text{iid}}{\sim} \text{Laplace}(R\epsilon^{-1})$.

**Theorem 3.4** (Informal version of Theorem D.2). *Let $S$ be an $\epsilon$-differentially private distribution sketch of a dataset $\mathcal{D} = \{x_1, ...x_N\}$ with $\mathcal{D} \underset{\text{iid}}{\sim} P(x)$ and let $k(x,y)$ be an LSH kernel (see Definition 3.2). Let $S$ define the function implemented by Algorithm 4. Then*

$$S(x) \underset{\text{i.p.}}{\to} \frac{1}{N} \sum_{x_i \in \mathcal{D}} k(x_i, x)$$

*with convergence rate $O(\sqrt{\log R / R} + \sqrt{R \log R}/(N\epsilon))$ when $R = \omega(1)$, e.g. $R = \log N$.*

In exchange for privacy, we pay a price in the convergence rate, which now has an additional $\epsilon^{-1}$ term. While this dependence can be improved to $\epsilon^{-1/2}$ (by optimizing $R$ for the fastest convergence), this requires *linear* memory, which is prohibitive in distributed settings.

### 3.2.2 Divergence Estimation

Under the conditions assumed by our analysis (continuous density on $\mathbb{R}^d$), the TVD between KDEs is a consistent estimator of the TVD [49, p1579]. Therefore, we can construct a consistent estimate of the TVD by simply plugging our estimators into Equation 3. The integral itself can be calculated using standard numerical integration techniques (e.g., Monte Carlo estimation or sparse grids), giving us a consistent sketch-based estimator for the TVD.

This method is implementable and comes with error bounds, but it requires a costly numerical integration whose complexity scales poorly with dimension ($O(d)$ per sample). Therefore, we justify a simpler method of computing the distance, albeit without formal guarantees.

As a collection of histograms, our sketch is a piecewise-constant approximation to the KDE over a partition $\pi$ of events. This permits a simpler expression for Equation 3, where we compute the value in each partition, weighed by the partition size. Here, we suppose that the partition $\pi$ consists of *cells* $\{A_1, ...\}$. We let $x \in A$ be any point inside $A$ and use $\lambda(A)$ to denote the Lebesgue measure of $A$.

$$\text{TVD}(p, q) = \frac{1}{2} \lim_{\max \lambda(A) \to 0} \sum_{x \in A \in \pi} |S_p(x) - S_q(x)| \lambda(A)$$

To further simplify this expression, we make the following observations. Each cell of the partition, which we denote as $A \in \pi$, is defined by the intersected buckets of the hash function. In other words, each cell contains the points that have the same hash mapping under all $R$ partitions.

While the expression for $\max_\pi \lambda(A)$ is not generally known for most LSH functions, it is clear that $\lambda(A)$ is non-increasing with $R$ because each cell of the partition $\pi$ is formed by intersecting $R$ sub-partitions (one from each hash function). This operation almost surely decreases the size of $A$ with each additional row in the sketch, since drawing two identical LSH functions is a zero-probability event. Also, $\mathbb{E}[\lambda(A)]$ is the same for all the cells in the partition by symmetry arguments.

Therefore, it is reasonable to consider an approximation where we compute entry-wise differences between sketches. Let $\Delta_{r,b}$ be the difference between row $r$ and column (or bucket) $b$ of the matrices $S_p$ and $S_q$. We can construct a TVD estimator as a sum over all possible intersections of hash buckets.

$$\text{TVD}(p, q) \approx B^{-R} \sum_{b_1=0}^{B} ... \sum_{b_R=0}^{B} |\Delta_{1,b_1} + ... + \Delta_{R,b_R}|$$

There are $B$ buckets in each repetition and, therefore, $B^R$ total "intersection paths" through the sketch. To circumvent the $O(B^R)$ complexity of computing each path, we consider the sum of absolute deviations $|\Delta_{1,b_1}| + ... + |\Delta_{R,b_R}|$ instead of $|\Delta_{1,b_1} + ... + \Delta_{R,b_R}|$. This gives an upper bound on the TVD that can be computed in $O(BR)$ time because we can now distribute the sums. In particular, we can compute the upper bound by taking the Frobenius norm (Euclidean distance) of $S_p - S_q$. We denote $\|S_p - S_q\|_2$ as *sketch distance*. Our empirical results show that this heuristic is an excellent indicator of distribution similarity.

## 4 One-Pass Distribution Sketch for FL

In this section, we introduce how to use our one-pass distribution sketch for effective and efficient FL. We will start with the client selection task with our sketch. Next, we will introduce how to perform an efficient cold start with our sketch in personalized FL shown in Section 6.

### 4.1 Distribution-Aware Client Selection

In our work, we study the setting where there are $n$ clients checking into the server. In every round of iterative federated optimization, $L$ clients are activated and ready for training. We assume that $L$ clients are activated uniformly random out of $n$ clients. Our client selection algorithm is presented in Algorithm 3 in the supplementary material. For each one of the $n$ clients, we compute its sketch with Algorithm 1 and send it to the server. As introduced in Theorem 3.4, our sketch can be $\epsilon$-differentially private with respect to the client data. As a result, we are able to communicate with the sketch between the server and clients. Next, the server takes the average of all the client sketches towards a global sketch $S_g$. In each round of iterative federated optimization, the server looks up the sketch for every activated client. Next, we compute the Euclidean distance between $S_g$ and every activated client sketch $S_j$. Next, we use the reciprocal of the distance as sample probability $p_j$. Next, we normalize the $p_j$ across the $L$ clients with Softmax function [50]. Next, we sample $K$ clients out of the $L$ activated clients. Each client is sampled with normalized probability $p_j$. Finally, we perform local training on the $K$ selected clients and aggregate their model updates.

---
**Algorithm 2** Cold Start with Distribution Sketch
---
**Input:** Clients $\mathcal{C} = \{c_1, \cdots, c_n\}$, New Client $C_q$, LSH function family $\mathcal{H}$, Hash range $B$, Rows $R$, Random seed $s$.
**Output:** Personalized model $w$.
**for** $i \in [n]$ **do**
    Get client data $\mathcal{D}_i$ from client $c_i$.
    Get client model $w_i$ from client $c_i$.
    Compute sketch $S_i$ using Algorithm 1 with parameter $\mathcal{D}_i$, $\mathcal{H}$, $B$, $R$ and $s$.
    Send $S_i$ to server
**end for**
Get client data $\mathcal{D}_q$ from client $c_q$.
Compute sketch $S_q$ using Algorithm 1 with parameter $\mathcal{D}_q$, $\mathcal{H}$, $B$, $R$ and $s$.
Send $S_q$ to the server.
$d_{\min} = \infty$
id = null
**for** $i \in [n]$ **do**
    $d_i = \|S_q - S_i\|_2$
    **if** $d_i < d_{\min}$ **then**
        $d_{\min} \leftarrow d_i$
        id $\leftarrow i$
    **end if**
**end for**
**return** $w_{\text{id}}$

---

Our algorithm has three features: (1) We propose a way of estimating global data distribution across clients without data exchange. (2) We can select clients with data distribution closer to the global distribution for a better global model. (3) Our algorithm is efficient. We only need one-shot, one-pass computation to generate a one-pass distribution sketch of client data. Moreover, the sketch is mergeable so a linear combination will give us a sketch for global data distribution. Furthermore, we can efficiently estimate the divergence of the client data distribution to the global distribution by taking the Euclidean distance of their sketches.

### 4.2 Nearest Neighbor Retrieval for Cold Start Client in Personalized FL

In this paper, we study the cold start problem in personalized FL. In this problem, many clients are connected to the server. Moreover, they collaborate together and produce a global model through iterative federated optimization [12]. Next, each client builds a personalized model based on the global model that achieves decent predictive performance on its local data (See Section 6). Next, a new client connects to the server. The new client has local but unlabeled data. In this setting, it

remains unknown which model, either global or local, should be applied to the new client as a cold starter.

We propose Algorithm 2 for this cold start problem. For the clients that already connect to the server, we compute its sketch with Algorithm 1 and send it to the server. Next, given a new client, we perform the sketching via Algorithm 1 with the same parameters for existing clients. Next, we compute the Euclidean distance between the new client sketch with the existing clients' sketches and retrieve the nearest neighbor client with minimum distance. Finally, the server retrieves the personalized model on the nearest neighbor client and sends it to the new client.

We want to highlight that Algorithm 2 formulates the cold start problem in personalized FL as a nearest neighbor search problem. By measuring data heterogeneity via Euclidean distance between distribution sketches, we provide an intuitive way for selecting a personalized client model.

# 5 Experiment

In this section, we conduct empirical evaluations to answer three questions: (1) Does the one-pass sketch distance reflect the differences between distributions? (2) Does the sketch distance help convergence in FL, and (3) Does the sketch distance retrieve the best-personalized models? To answer these three questions, we conducted three sets of experiments.

## 5.1 Dataset, Model and Implementation

**Dataset:** We evaluate Algorithm 3 and Algorithm 2 on both vision and language datasets. For visual classification, we use the MNIST dataset [51] and FEMNIST [52]. For MNIST, we introduce a federated dataset where each client is generated by random sampling from MNIST under Dirichlet distribution with $\alpha = 0.5$. We also introduce *MNIST Uniform + Dirichlet*, a dataset with half of the client data generated from random sampling from MNIST under Dirichlet distribution with $\alpha = 0.5$ and the other half of the client generated from uniform sampling from MNIST. We also use the Shakespeare next-character prediction dataset [6] for language-based FL. The clients in both FEMNIST and Shakespeare datasets are fixed.

**Model:** For the MNIST dataset, we use a fully-connected network with a single 512-dimension hidden layer. For the FEMNIST dataset, we use LeNet-5 convolutional network [53, 51]. For the Shakespeare dataset, we build a model with one embedding layer, one long short-term memory (LSTM) [54] block, and an output layer. The input and output vocabulary size is 80, and the hidden dimension of the model is 256.

**Implementation:** Our FL codebase, including FL workflow, LSH functions, and proposed algorithms, is implemented on PyTorch [55]. We test Algorithm 3 and Algorithm 2 on a server with 8 Nvidia Tesla V100 GPU and a 48-core/96-thread processor (Intel(R) Xeon(R) Gold 5220R CPU @ 2.20GHz).

## 5.2 One-Pass Sketch Distance

In this section, we investigate the correlation between one-pass sketch distance and distribution differences.

**Setting:** We generate two groups of subsets from MNIST, Group A and Group B. Datasets from Group A are drawn randomly from MNIST under Dirichlet distribution with $\alpha = 0.5$. Datasets from Group B are drawn randomly from MNIST under the uniform distribution. For every dataset, we construct a one-pass sketch. Then, we calculate the pairwise one-pass sketch distances between every dataset.

**Result:** We present the result in Figure 1. We annotate the one-pass sketch distance in each cell. First, we observe symmetry along the diagonal, which satisfies the basic principle of distance, $d(x, y) = d(y, x)$. Second, we observe group separation. The pairwise distances within group B are smaller, which is expected as they were drawn uniformly. The pairwise distances within Group A are larger and more diverse, which is expected as each is drawn randomly from the Dirichlet distribution. Lastly, for any dataset from Group A, its distance from any dataset from Group B is similar. Our results answer the first question: the distance between the distribution sketches constructed by Algorithm 1 can reflect their divergence in the distribution.

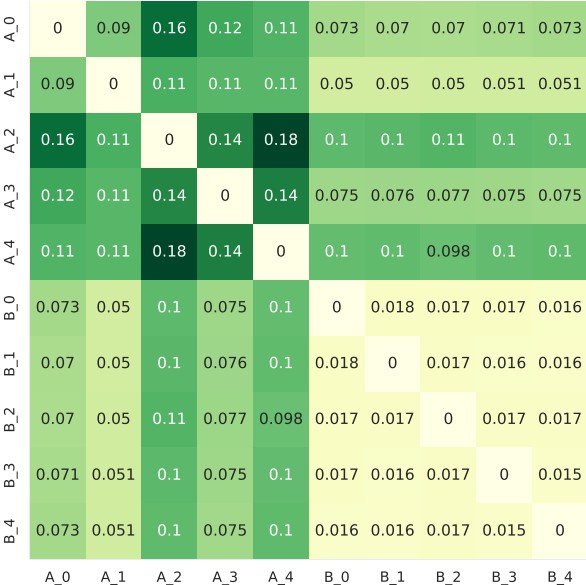

Figure 1: Pairwise one-pass sketch distance visualization. $A_0$, $A_1$, ... $A_n$ are five datasets drawn randomly under the Dirichlet distribution. $B_0$, $B_1$, ... $B_n$ are five datasets drawn randomly under the uniform distribution. We mark each cell with the one-pass sketch distance shown in Algorithm 1.

## 5.3 Distribution-Aware Client Sampling

In this section, we apply the one-pass distribution sketch (see Algorithm 1) in the client selection setting (See Section 6) and investigate whether data heterogeneity measurement helps with FL towards faster convergence. **Setting:** As shown in Section 4.2, in every round of iterative federated

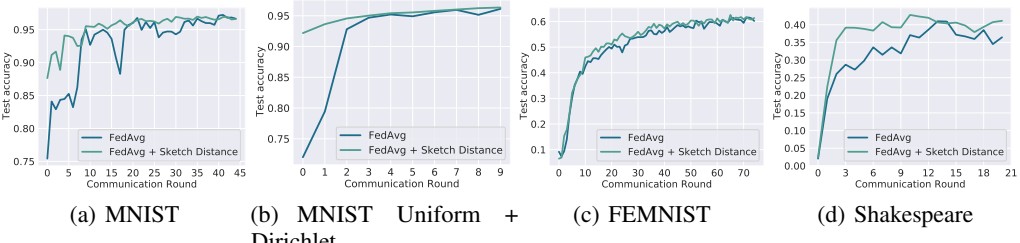

|                     |                     |                     |                     |
|---------------------|---------------------|---------------------|---------------------|
| (a) MNIST           | (b) MNIST Uniform + Dirichlet | (c) FEMNIST | (d) Shakespeare |

Figure 2: Result: Convergence plot for MNIST, MNIST Uniform+Dirichlet, FEMNIST, and Shakespeare.

optimization, there would be $L$ activated clients. The standard federated optimization approach would uniformly randomly sample $K$ clients as participants [6]. We denote this approach as *Fedavg*. However, due to the data heterogeneity, we often observe model divergence during aggregation. As a result, we propose Algorithm 3 to select the clients with distribution closer to the global distribution. In this experiment, we compare Algorithm 3 with Fedavg to showcase the importance of data heterogeneity in client selection. We set $L = 3K$. For the MNIST dataset (both MNIST and MNIST Uniform + Direchlet), both Algorithm 3 and Fedavg are trained by 200 rounds. In each round, $K = 3$ clients are selected from $L$ active clients. Next, each client is trained for 20 epochs with batch size 32 and learning rate $\eta = 0.0001$. The FEMNIST uses the same training rounds, epochs, batch size, and learning rate as MNIST, except $K = 10$. For the Shakespeare dataset, both Algorithm 3 and Fedavg are trained by 20 rounds. Each round, $K = 3$ clients are selected from $L$ active clients. Next, each client is trained for 20 epochs with batch size 10 and learning rate $\eta = 0.01$. We introduce early stopping to prevent overfitting.

**Result:** We present the test accuracy plot in Figure 2. As shown in the figure, on four FL datasets/settings, Algorithm 3 converges faster than Fedavg. In almost every round of optimization,

Table 1: This table summarizes the test accuracy result for the cold client setting.

| Dataset | Warm Client | | Cold Client | |
|---|---|---|---|---|
| | Global Model | Local Model | Global Model | Neighbor Model |
| MNIST | $0.8764 \pm 0.0084$ | $0.9784 \pm 0.0016$ | $0.8791 \pm 0.0136$ | $0.9702 \pm 0.0004$ |
| FEMNIST | $0.6593 \pm 0.0039$ | $0.7336 \pm 0.0064$ | $0.6667 \pm 0.0044$ | $0.6937 \pm 0.0117$ |
| Shakespeare | $0.3581 \pm 0.0250$ | $0.3645 \pm 0.0249$ | $0.3237 \pm 0.0186$ | $0.3455 \pm 0.0082$ |

Algorithm 3 achieves better accuracy than Fedavg. This experiment's results answer the second question. The distances between distribution sketches can be used to sample helpful clients towards faster convergence.

## 5.4 Cold Start in Personalized FL

In this section, we would like to evaluate the performance of Algorithm 2 in the cold start problem of personalized FL (see Section 4.2).

**Setting:** For every dataset, we uniformly randomly divided them into warm clients and cold clients. The warm clients will participate in global training using Fedavg. Once we have trained the global model. Each client locally fine-tunes the global model. As a result, we have a global model on the server, and each client has its personalized model. For cold clients, we only use its label for evaluation. We use MNIST, FEMNIST, and Shakespeare datasets as introduced in Section 5.1. We generate MNIST clients following the by-class format shown in [13]. There are 100 clients in total. We set half of them as cold clients and the rest of them as warm clients. For MNIST, to get the global model, we train 200 rounds, and each round uniformly selects 10 clients. Each client is trained for 10 epochs on its local data with batch size 32 and learning rate 0.0001. Next, we finetune each client for 10 epochs with the same batch size and

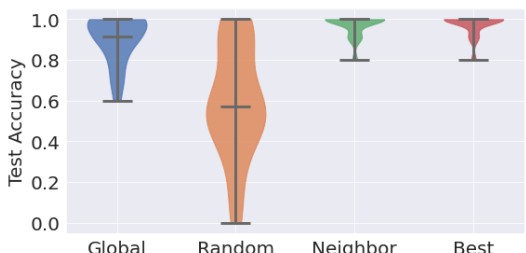

Figure 3: Test accuracy of cold clients for MNIST. **Global**: the model trained by Fedavg, **Random**: uniformly random sample a personalized client model, **Neighbor**: model selected by Algorithm 2, **Best**: the upper bound of test accuracy.

learning rate global model training for a local model. For FEMNIST, we set half of the clients as cold clients and the rest as warm clients. We use the same parameters as MNIST, except each round uses 50 clients. For Shakespeare, we use 20 clients as cold start clients, and the rest of them are warm clients. To get the global model, we train 5 rounds, and each round uniformly selects 3 clients. Each client is trained for 20 epochs on its local data with batch size 10 and learning rate 0.01. Next, we finetune each client for 20 epochs for a local model. We set the batch size and learning rate as the same as global model training.

**Result:** We present the result in Table 5.4. For warm clients, we observe that the local models generally have better test accuracy due to local finetuning. For cold clients, using the model from its nearest neighbor always achieves high accuracy compared to the global model. Moreover, we provide an ablation study on MNIST. We obtain the test accuracy on each cold client with the global model, the model retrieved by uniformly random sampling, and the model retrieved by Algorithm 2. We also run a brutal force search on the existing personalized models and find the best model with the highest test accuracy. Note that this search is impractical since we use client data label information. But it is an upper bound for Algorithm 2. As shown in Figure 3, we observe that Algorithm 2 outperforms both the global model and random sampling with higher test accuracy on clients. Moreover, Algorithm 2 approximates the upper bound well. The experiment in this section answers the third question: Algorithm 2 can retrieve suitable personalized models for new clients with unlabeled data.

## 6 Related Work

**Client Selection in FL.** Iterative federated optimization is the de-facto standard for FL. In this optimization strategy, a server aggregates the model updates from each device to create a better

global model. Client selection [56, 57, 58, 59] serves as an essential component of FL in the common situation where many clients are connected to the server. In each round of iterative federated optimization, the server identifies a subset of clients as participants [60], asks participants to train on client data, and finally aggregates the updates. Client selection is necessary for two reasons. First, large-cohort training can cause optimization algorithms to fail. If we use all the clients, local objective misalignment can lead to catastrophic training failures [61]. This observation is backed up by theoretical analysis, which suggests that proper client selection can stabilize convergence [13, 62, 63]. Second, there are systems challenges. The communication required to coordinate a large number of clients imposes a heavy burden on network bandwidth [64, 65], reducing the efficiency of the FL system. Recent work on client selection strategies in FL focuses on the model's performance on the client data. For instance, [66] and [67] use the training loss of the client model on its data as a utility score for client selection. [68] and [69] use client model updates and build importance scores to select clients. In this work, we focus on using data heterogeneity for client selection without model information. As a result, we can avoid the computation and communication on unselected clients.

**Personalized FL.** Personalization is a natural way of improving local predictive performance on a client [70, 71, 27, 72]. Given the global model trained across diverse clients, it is a common practice to fine-tune the global model on every client's data to obtain a better local model [14, 73]. In this paper, we focus on a new challenging setting from practice: the cold start problem in personalized FL. In this setting, there are plenty of clients with personalized models on their devices. Next, a new client comes with unlabeled data. We want to cold start this client with a suitable local model. In this way, we can still provide decent performance on this client without training. Our work focuses on retrieving a local model from a client with a similar data distribution to the new client. We aim to achieve this strategy with an effective data heterogeneity measurement.

## 7 Conclusion

We study the problem of data heterogeneity in federated learning (FL), where different clients contain data generated from different distributions. We argue that data heterogeneity should be effectively measured for better global model training and FL personalization. However, there is no explicit and efficient way of measuring data heterogeneity. To address this challenge, we propose a one-pass distribution sketch that represents the data distribution while preserving $\epsilon$-differential privacy. We show that the sketch distance can approximate the total variation distance between clients and introduce this technique to the client selection and cold start personalization tasks in FL. Our experiments showcase our sketch's practical impact in effective and efficient FL.

## 8 Acknowledgement

We would like to thank Tian Li, Jianyu Wang, and the anonymous reviewers for helpful discussions and feedback. Zichang Liu, Benjamin Coleman, and Anshumali Shrivastava are supported by NSF-CCS-2211815, ONR-DURIP and NSF-BIGDATA-1838177. Zhaozhuo Xu is supported by startup funding from the Stevens Institute of Technology.

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

# Appendix

## A  Design of LSH Functions:

In practice, we could realize $d(x, y)$ with cosine similarity for dense vectors. In this case, the LSH function family $\mathcal{H}$ should have the following form.

**Definition A.1** (Sign Random Projection (SRP) Hash [74]). *We define a function family $\mathcal{H}$ as follow: Given a vectors $x \in \mathbb{R}^d$, any $h : \mathbb{R}^d \to \{0, 1\}$ that is in the family $\mathcal{H}$, we have*

$$h(x) = \mathsf{sign}(Ax),$$

*where $A \in \mathbb{R}^{d \times 1}$ is a random matrix that every entry of $A$ is sampled from normal distribution $\mathcal{N}(0, 1)$, $\mathsf{sign}$ is the sign function that set positive value to $1$ and others to $0$. Moreover, for two vectors $x, y \in \mathbb{R}^d$ we have*

$$\Pr[h(x) = h(y)] = 1 - \pi^{-1} \arccos \frac{x^\top y}{\|x\|_2 \|y\|_2}.$$

As shown in Definition A.1, the SRP hash is an LSH function built upon random Gaussian projections, which is a fundamental technique in ML [75, 76, 77] If two vectors are close in terms of cosine similarity, their collision probability would also be high.

The SRP hash is usually designed for dense vectors. If both $x \in \{0, 1\}^d$ and $y \in \{0, 1\}^d$ are high dimensional binary vectors with large $d$. Their Jaccard similarity [78] is also an important measure for search [79, 80, 81] and learning tasks [82, 83]. There also exists a family of LSH functions for Jacacrd similarity. We define this LSH function as:

**Definition A.2** (MinHash [84]). *A function family $\mathcal{H}$ is a MinHash family if for any $h \in \mathcal{H}$, given a vectors $x \in \{0, 1\}^d$, we have*

$$h(x) = \arg\min(\Pi(x))$$

*where $\Pi$ is a permutation function on the binary vector $x$. The $\arg\min$ operation takes the index of the first non-zero value in $\Pi(x)$. Moreover, given two binary vectors $x, y \in \{0, 1\}^d$, we have*

$$\Pr[h(x) = h(y)] = \frac{\sum_{i=1}^d \min(x_i, y_i)}{\sum_{i=1}^d \max(x_i, y_i)},$$

*where the right term represents the Jaccard similarity of binary vectors $x$ and $y$.*

Following Definition A.2, MinHash serves as a powerful tool for Jaccard similarity estimation [85, 86, 87]. We will use both SRP hash and MinHash in the following section to build a sketch for data distribution.

In this paper, we take a kernel view of the collision probability of LSH (see Definition 3.2). This view aligns with a series of research in efficient kernel decomposition [88, 89, 90], kernel density estimation [46] and kernel learning [91].

## B  More Algorithms

In this section, we introduce the client selection algorithm with our one-pass sketch.

**Algorithm 3** Client Selection with Distribution Sketch
***

**Input:** Clients $\mathcal{C} = \{c_1, \cdots, c_n\}$, Number of rounds $T$, Step size $\eta$, LSH function family $\mathcal{H}$, Hash range $B$, Rows $R$, Number of active clients $L$, Number of Selected Clients $K$, Epochs $E$, Random Seed $s$
**Output:** Global model $w^T$.
**Initialize:** Global model $w^1$, global sketch $S_g$, random seed $s$.
**for** $c \in \mathcal{C}$ **do**
    Get client data $\mathcal{D}_c$ from client $c$.
    Compute sketch $S_c$ using Algorithm 1 with parameter $\mathcal{D}_c$, $\mathcal{H}$, $B$, $R$ and $s$.
    Send $S_c$ to server
    $S_g \leftarrow S_g + S_c$
**end for**
$S_g \leftarrow S_g/n$                                                   ▷ Generate global sketch on server
**for** $i \in [T]$ **do**
    $L$ clients $\{c_1, \cdots, c_L\} \subset \mathcal{C}$ are activated at random.
    **for** $j \in [L]$ **do**
        $p_j = 1/\|S_g - S_j\|_2$                              ▷ $S_j$ is the sketch for client $c_j$
    **end for**
    **for** $j \in [L]$ **do**
        $p_j = \frac{\exp p_j}{\sum_{l=1}^{L} \exp p_l}$
    **end for**
    Sample $K$ clients out of $\{c_1, \cdots, c_L\}$ without replacement, client $c_j$ is selected with probability $p_j$.
    Server send $w^i$ to the selected clients.
    Each selected client $c_k$ updates $w^i$ to $w_k^i$ by training on its data for $E$ epochs with step size $\eta$.
    Each selected client sends $w_k^i$ back to the server.
    $w^{i+1} = \sum_{k=1}^{K} w_k^i$
**end for**
**return** $w^T$
***

# C   More Experiments

In this section, we provide two-fold of more experiments. We start with a simulation of our sketch (see Algorithm 1) for total variation distance (TVD) estimation. Next, we perform client selection (see Algorithm 3) on CIFAR-10 datasets to observe the convergence improvements of our approach over FedAvg.

## C.1   Simulation for Total Variation Distance Estimation

We computed the total variation distance (TVD) for two Gaussians with the same covariance and different means, one of the few TVDs between high-dimensional distributions for which we have an exact formula. This formula is $\text{TVD} = 2\Phi(z) - 1$ where $z = \sqrt{(\mu_1 - \mu_2)^\top C^{-1}(\mu_1 - \mu_2)}$ (see [34, 92] for reference). To get the value of $\lambda(A)$ (see Section 3.2.2), we integrated the high-dimensional collision probability over the d-ball. We drew 1000 samples from two 16-dimensional Gaussians with random means, computed their sketch and TVD distances, and compared the results.

## C.2   CIFAR-10 Experiments

We conducted CIFAR-10 experiments using 100 clients, where L=3K and K=2. The client dataset follows a non-iid distribution, as specified in the split provided at `https://github.com/yjlee22/FedShare`. We visualized the iteration-wise convergence of FedAvg with and without the proposed client selection (see Figure 5). The results demonstrate that our distribution sketch improves the test accuracy convergence rate, leading to faster iterations.

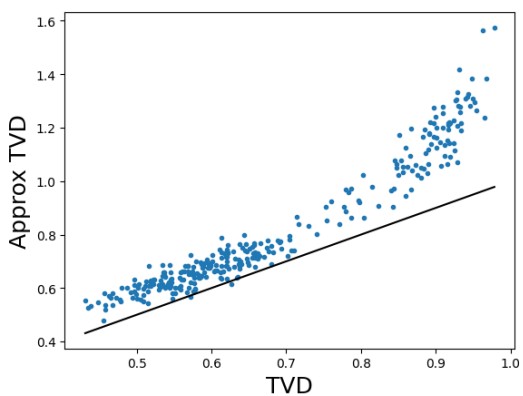

Figure 4: Visualization of TVD Estimation via Sketch.

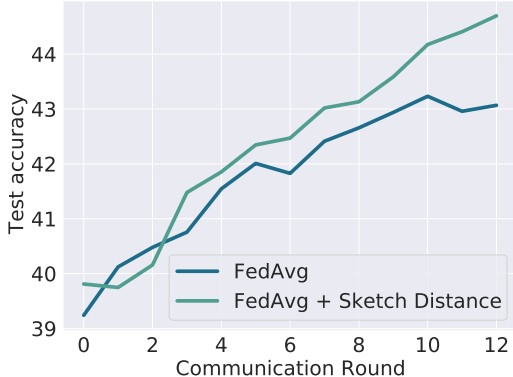

Figure 5: CIFAR-10 Experiments: We visualized the iteration-wise convergence of FedAvg with and without the proposed client selection.

## D   Proofs

To formally prove the Theorems in the paper. We start with introducing a query algorithm to the one-pass distribution sketch.

---

**Algorithm 4** Query to the Distribution Sketch

---

**Input:** Query $q \in \mathbb{R}^d$, LSH functions $h_1, \ldots, h_R$, Dataset $\mathcal{D}$, Sketch $S \in \mathbb{R}^{R \times B}$ built on $\mathcal{D}$ with Algorithm 1
**Output:** $S(q) \in R$
**Initialize:** $S(q) \leftarrow 0$
$A \leftarrow \emptyset$
**for** $i = 1 \rightarrow R$ **do**
    $A \leftarrow A \cup \{S_{i,h_i(q)}\}$
**end for**
$S(q) \leftarrow \mathsf{median}(A)$
**return** $S(q)$

---

Next, we introduce the formal statements in the paper as below.

**Theorem D.1** (Formal version of Theorem 3.3). *Let $p(x)$ denote a probability density function. Let $\mathcal{D} \underset{\mathrm{iid}}{\sim} p(x)$ denote a dataset. Let $k(x, y)$ be an LSH kernel (see Definition 3.2). Let $S$ define the*

*function implemented by Algorithm 4. We show that*

$$S(x) \underset{\text{i.p.}}{\to} \frac{1}{N} \sum_{x_i \in \mathcal{D}} k(x_i, q)$$

*with convergence rate $O(\sqrt{\log R}/\sqrt{R})$.*

*Proof.* Let $\kappa(x) = \sum_{x_i \in \mathcal{D}} \sqrt{k(x, x_i)}$ be the (non-normalized) kernel density estimate. Theorem 3.4 of [48] provides the following inequality for any $\delta$, where $\widetilde{\kappa}(x) = \sum_{x_i \in \mathcal{D}} \sqrt{k(x, x_i)}$:

$$\Pr \left[ |NS(x) - \kappa(x)| > \left( 32 \frac{\widetilde{\kappa}^2(x)}{R} \log 1/\delta \right)^{1/2} \right] < \delta$$

This is equivalent to the following inequality, which we can obtain by dividing both sides of the inequality inside the probability by $N$.

$$\Pr \left[ \left| S(x) - \frac{1}{N} \kappa(x) \right| > \left( 32 \frac{\widetilde{\kappa}^2(x)}{N^2 R} \log 1/\delta \right)^{1/2} \right] < \delta$$

We want to show that the error $\left| S(x) - \frac{1}{N} \kappa(x) \right|$ converges in probability to zero, because this directly proves the main claim of the theorem. To do this, we must show that for any $\Delta > 0$,

$$\lim_{R \to \infty} \Pr \left[ \left| S(x) - \frac{1}{N} \kappa(x) \right| > \Delta \right] = 0$$

This can be done by setting $\delta = \frac{1}{R}$, which yields the following inequality:

$$\Pr \left[ \left| S(x) - \frac{1}{N} \kappa(x) \right| > \left( 32 \frac{\widetilde{\kappa}^2(x)}{N^2 R} \log R \right)^{\frac{1}{2}} \right] < \frac{1}{R}$$

Because $\widetilde{\kappa} < N$, the following (simpler, but somewhat looser) inequality also holds:

$$\Pr \left[ \left| S(x) - \frac{1}{N} \kappa(x) \right| > \left( 32 \frac{\log R}{R} \right)^{\frac{1}{2}} \right] < \frac{1}{R}$$

This implies that $S(x) \underset{\text{i.p.}}{\to} \frac{1}{N} \kappa(x)$ by considering $R$ large enough that $\sqrt{\log R/R} < \Delta$. $\qquad \square$

**Theorem D.2** (Formal version of Theorem 3.4). *Let $S$ be an $\epsilon$-differentially private distribution sketch of a dataset $\mathcal{D} = \{x_1, ...x_N\}$ with $\mathcal{D} \underset{\text{iid}}{\sim} p(x)$ and let $k(x, y)$ be an LSH kernel (see Definition 3.2). Let $S$ define the function implemented by Algorithm 4. Then*

$$S(x) \underset{\text{i.p.}}{\to} \frac{1}{N} \sum_{x_i \in \mathcal{D}} k(x_i, x)$$

*with convergence rate $O(\sqrt{\log R/R} + \sqrt{R \log R}/(N\epsilon)$ when $R = \omega(1)$ (e.g. $R = \log N$).*

*Proof.* As before, let $\kappa(x) = \sum_{x_i \in \mathcal{D}} \sqrt{k(x, x_i)}$ be the (non-normalized) kernel density estimate and let $\widetilde{\kappa}(x) = \sum_{x_i \in \mathcal{D}} \sqrt{k(x, x_i)}$. For the $\epsilon$-differentially private version of the algorithm, Theorem 3.4 of [48] provides the following inequality for any $\delta > 0$.

$$\Pr \left[ |NS(x) - \kappa(x)| > \left( \left( \frac{\widetilde{\kappa}^2(x)}{R} + 2 \frac{R}{\epsilon^2} \right) 32 \log 1/\delta \right)^{1/2} \right] < \delta$$

Again, we divide both sides of the inner inequality by $N$, and we also loosen (and simplify) the inequality with the observation that $\widetilde{\kappa}(x)/N < 1$.

$$\Pr \left[ \left| S(x) - \frac{1}{N} \kappa(x) \right| > \left( \left( \frac{1}{R} + 2 \frac{R}{N^2 \epsilon^2} \right) 32 \log 1/\delta \right)^{1/2} \right] < \delta$$

We want to show that the error $\left|S(x) - \frac{1}{N}\kappa(x)\right|$ converges in probability to zero, because this directly proves the main claim of the theorem. To do this, we must show that for any $\Delta > 0$,

$$\lim_{R\to\infty} \Pr\left[\left|S(x) - \frac{1}{N}\kappa(x)\right| > \Delta\right] = 0$$

As before, we choose $\delta = \frac{1}{R}$, which yields the following inequality:

$$\Pr\left[\left|S(x) - \frac{1}{N}\kappa(x)\right| > \left(\left(\frac{1}{R} + 2\frac{R}{N^2\epsilon^2}\right)32\log R\right)^{1/2}\right] < \frac{1}{R}$$

This is the same as:

$$\Pr\left[\left|S(x) - \frac{1}{N}\kappa(x)\right| > \left(32\frac{\log R}{R} + 64\frac{R}{N^2\epsilon^2}\log R\right)^{1/2}\right] < \frac{1}{R}$$

Here, we will loosen the inequality again (also for the sake of presentation). Because $\sqrt{a+b} \leq \sqrt{a} + \sqrt{b}$, the following inequality is also satisfied:

$$\Pr\left[\left|S(x) - \frac{1}{N}\kappa(x)\right| > 4\sqrt{2}\sqrt{\frac{\log R}{R}} + 8\frac{\sqrt{R\log R}}{N\epsilon}\right] < \frac{1}{R}$$

Here, the convergence rate is $O(\sqrt{\log R/R} + \sqrt{R\log R}/(N\epsilon))$. For us to have the error converge in probability to zero, we need for

$$\sqrt{\frac{\log R}{R}} + \sqrt{2}\frac{\sqrt{R\log R}}{N\epsilon} \to 0$$

as $N \to \infty$. A simple way to achieve this is for $R$ to be weakly dependent on $N$ (i.e. choose $R = \omega(1)$). For example, choosing $R = \log N$ satisfies the conditions. $\square$

