# OpenReview forum: "One-Pass Distribution Sketch for Measuring Data Heterogeneity in Federated Learning"
_NeurIPS.cc/2023/Conference — NeurIPS 2023 poster_

### Official Review · Reviewer_c2Su · 2023-07-03

**Soundness:** 3 good
**Presentation:** 3 good
**Contribution:** 3 good
**Rating:** 7
**Confidence:** 4

**Summary:**

The paper proposes to use a locally sensitive hashing based sketch to measure the similarity between two datasets in a privacy-preserving manner. This sketch can be computed efficiently and differential privacy can be guaranteed via the Laplace mechanism. The paper concludes with presenting two potential applications of their similarity measure: selecting participating clients in federated learning based on the similarity of local datasets, and selecting an initial model for a new client in personalized federated learning.

**Strengths:**

- The idea is clear and sound, a privacy-preserving dataset similartiy measure is useful, in particular in federated and collaborative learning scenarios.
- Theoretically sound definition of the sketch from which an estimator of the total variation distance can be derived.
- The paper presents two applications of the similarity measure for federated learning.

**Weaknesses:**

- The distribution-aware client sampling is interesting, but not well-motivated. Sampling clients with differing local data from the average distribution can be helpful in some settings, while being detrimental to the learning process in others. While the empricial results look promising, the analysis is far too shallow to permit conclusions.
--- after rebuttal ---
The authors have addressed this weakness in the rebuttal.

**Questions:**

- Would the definition 3.1 of TVD not be more appropriately placed at the beginning of Sec 3.2.2?
- Would it be possible to quantify the difference of the upper bound (using the Euklidean distance between sketches) to the true TVD, based on properties of the sketches? E.g., assuming sufficiently smooth data distributions.
- Could you add synthetic experiments where data is drawn from known distributions, so that the accuracy of the TVD estimation can be more clearly assessed?

**Limitations:**

The paper is clear about the assumptions and limitations.

---

> ### Author Rebuttal · Authors · 2023-08-09
>
>
> We thank the reviewer for the comments and suggestions to improve our paper! Please see the following clarifications.
>
> **Q: Would the definition 3.1 of TVD not be more appropriately placed at the beginning of Sec 3.2.2?**
>
> We introduce the TVD early to motivate the theory and provide a high-level sketch of the ideas. However, we agree that equation (2) would be better placed in Section 3.2.2 and plan to clean this up in the revision.
>
> **Q: Would it be possible to quantify the difference of the upper bound (using the Euklidean distance between sketches) to the true TVD, based on properties of the sketches? E.g., assuming sufficiently smooth data distributions.**
>
> This is difficult to do rigorously, even if we assume a sufficiently smooth data distribution. We can think of two ways to prove rigorous estimation bounds for the TVD, but both have technical difficulties. The first way is to analyze the sketch as a high-dimensional histogram and formalize the arguments of Section 3.2.2. However, we cannot calculate the error of the piecewise integral approximation without upper and lower bounds on $\lambda(A)$ (which are not generally known for LSH partitions). The second way is to analyze the sketch as a density estimator, and then apply a known technique for high-dimensional integration (such as Monte Carlo or sparse-grids). We briefly describe this option in the first paragraph of Section 3.2.2, but unfortunately we are not aware of a high-dimensional integration method that comes with the computational performance we require in this application.
>
> **Q: Could you add synthetic experiments where data is drawn from known distributions, so that the accuracy of the TVD estimation can be more clearly assessed?**
>
> This is an excellent suggestion. We computed the TVD for two Gaussians with the same covariance and different means, one of the few TVDs between high-dimensional distributions for which we have an exact formula. This formula is $\mathrm{TVD} = 2\Phi(z) - 1$ where $z = \sqrt{\mu_1 - \mu_2)^{\top}C^{-1}(\mu_1 - \mu_2)}$ (see Theorem 1 of [1] and page 5 of [2]). To get the value of $\lambda(A)$ from line 178, we integrated the high-dimensional collision probability over the d-ball. We drew 1000 samples from two 16-dimensional Gaussians with random means, computed their sketch and TVD distances, and compared the results. We plot the results in the attached PDF. We will add this discussion to the paper.
>
> [1]: "Estimates of the proximity of Gaussian measures," S. S. Barsov and V. V. Ulyanov, Soviet Mathematics, Doklady, 34:462–466, 1987.
>
> [2]: “The total variation distance between high-dimensional Gaussians with the same mean,” Devroye et. al., arXiv 2022.

---

> > ### Comment · Reviewer_c2Su · 2023-08-14
> > **Answer to rebuttal**
> >
> > I thank the authors for their answers, in particular the experiment on TVD estimation. It indicates that the proposed sketching technique is a reasonable estimator for TVD.
> >
> > I have seen that also an additional experiment on client selection on CIFAR10 has been performed. Would you be able to comment on the motivation of distribution-aware sampling in the light of those results? What I mean is: please discuss the scenarios in which distribution-aware sampling is reasonable, and where it is not (e.g., from the perspective of transfer learning). In the end, I just want to see a proper discussion of the assumptions and limitations for that use case.
> >
> > If that weakness is addressed, I think this is a good paper and I would increase my score.

---

> > > ### Author Response · Authors · 2023-08-16
> > > **Thank you for the encouragement**
> > >
> > > We appreciate the reviewer's encouragement! We agree that proper diction of the assumption and limitation is essential.
> > >
> > > We will address the distribution-aware client sampling and cold-start problem separately.
> > >
> > > Our client selection process (Algorithm 3) effectively acts as a reweighing of the objective function to focus on the head of the distribution and avoid overfitting on the tail. This improves the stability and overall head performance of the original model, though it does de-prioritize long-tail inputs. We expect this client selection process to benefit applications where the data is noisy and performance depends on correct prediction in the head. It should be noted that most recommendation applications fall into this category. It is standard practice for industrial recommender systems to focus on the highly shared parts of the input domain and to ignore / alias the tail [1]. On the other hand, this will be less beneficial for applications that rely on coverage over the whole space.  We've made updates to the paper, emphasizing the assumptions and limitations of our chosen sampling approach.
> > >
> > > Cold-start problem is reasonable when we expect clients to cluster into different groups with cluster-dependent behavior/distributions. For example, consider personalized recommendation models. In many recommendation applications, web traffic is divided into categories (e.g., by country or zip code), and an independent model is used within each category. Our cold-start algorithm automatically groups the clients to minimize the distribution mismatch, improving the performance (and reducing the required size/capacity) of each model. We expect this process to work whenever it is reasonable to perform transfer learning from a user to a group of similar users. Note that while our experiments focus on cold-start performance, it is entirely possible to train federated learning models within each grouping (thereby performing a variant of clustered federated learning, where the clusters are optimized for distribution match).
> > >
> > > [1]Categorical Feature Compression via Submodular Optimization

---

### Official Review · Reviewer_iXQ3 · 2023-07-04

**Soundness:** 2 fair
**Presentation:** 2 fair
**Contribution:** 2 fair
**Rating:** 6
**Confidence:** 3

**Summary:**

The presence of **data heterogeneity** poses a critical challenge in **federated learning** (FL), as it leads to slower and noisier collaborative learning. To address this issue and improve the learning process with faster convergence, it is here proposed a more efficient client selection strategy based on distribution estimation. By utilizing an **efficient distribution sketch**, the server gains access to estimated distributions of the clients' local data, enabling the construction of a sketch representing the underlying global distribution while maintaining privacy constraints. With this information at hand, clients with data distributions closer to the global one can be more actively involved in the training process, thereby mitigating the negative effects of data heterogeneity and facilitating the development of a better global model. In addition, this approach offers the advantage of providing new clients - potentially holding unlabeled data - with initial models that are better aligned with their local distribution, thus enhancing their local training. To be more specific, new clients can receive the model trained by other users with similar local distributions, thus benefiting from their previously acquired knowledge.

**Strengths:**

1. The paper addresses the issue of data heterogeneity in FL, relevant to this field of research.
2. A new method for estimating local and global data distributions is introduced. This method is both efficient and privacy preserving. It can also be adapted to be $\epsilon$-differentially private. The distribution sketch enables convergence speed up by allowing a smarter client selection, thus reducing the communication rounds between clients and server, significant for the evaluation of FL algorithms.
3. Both theoretical and complexity analyses are presented for the proposed method. Claims are supported by proofs.
4. Empirical validation is presented on both vision and NLP datasets.
5.The paper is well written, easy to follow and understand.
6. Limitations and potential negative impacts are discussed. Computational requirements are identified.

**Weaknesses:**

1. Some relevant related works are missing.
    * Client selection: [1,2]
    * Personalized FL: [3]
    * I believe discussing other works proposing different data distribution estimation techniques is relevant to this paper, but is currently missing. Examples are [4,5,6,7].
    * Algorithms addressing statistical heterogeneity in FL. Examples are unlimited: the most well-known baselines are [10-15]. Recently works such as [16,17] are gaining visibility, as well as [18].
    * Lastly, since the paper highlights the positive effect of having new clients starting from pretrained models, current literature addressing the impact of pretraining in FL might be of interest [8,9].
2. The efficacy of the one-pass distribution sketch in efficiently addressing the cold start of new *unlabeled* clients is emphasized. However, it is important to note that these clients still require labeling in order to effectively use the model and perform local training. As no alternative solution is proposed in this context, the necessity for labeled data on the client side remains unchanged. Consequently, the ability for new clients to initially possess unlabeled data does not yield a substantial difference in practical real-world scenarios.
3. In realistic cross-device [19] federated scenarios, billions of clients with very small datasets are involved in the training. This implies $|\cup_{k\in[K]} \mathcal{D}_k| \gg |\mathcal{D}_k|$, with $K$ being the number of clients with local datasets $\mathcal{D}_k, k\in[K]$. As a consequence, it is very unlikely that distribution of single local datasets may resemble the global one. Referring to the notation introduced in Algorithm 3, in the real-world scenario, $p_i - p_j \approx 0 \forall i,j\in[L]$. It is not clear how this method would be effective in this case.
4. To my understanding, in order to perform the warm start on new clients, the server has to keep memory of the last model trained by each client, and store it, together with the distribution sketch. This becomes unfeasible with the increasing number of clients in realistic settings. If the server retrieves the personalized model from the nearest neighbor client instead, this introduces additional communication, and such cost increases with the constant arrival of new clients. This limitation is nowhere addressed.
5. By favoring clients having local distributions similar to the global one, the resulting learned model might potentially be unfair towards out-of-distribution users. I believe this behavior arises issues in terms of **fairness** towards less represented distributions.
6. The method is only compared with FedAvg, the first algorithm proposed for FL. Missing comparison with methods i) using different techniques for estimating the local distributions, ii) performing smarter client selection, iii) addressing statistical heterogeneity in FL, iv) using pretrained models in FL.

**Minor weaknesses** (did not affect my rating)
1. Missing hyperparameters ranges used for tuning the baselines and some details on the federated settings. Please refer to Questions.

**References**

[1] Cho, Yae Jee, Jianyu Wang, and Gauri Joshi. "Towards understanding biased client selection in federated learning." International Conference on Artificial Intelligence and Statistics. PMLR, 2022.

[2] Yang, Wenkai, et al. "When to Trust Aggregated Gradients: Addressing Negative Client Sampling in Federated Learning." TMLR (2023).

[3] Fallah, Alireza, Aryan Mokhtari, and Asuman Ozdaglar. "Personalized federated learning with theoretical guarantees: A model-agnostic meta-learning approach." Advances in Neural Information Processing Systems 33 (2020): 3557-3568.

[4] Zeng, Shenglai, et al. "Heterogeneous federated learning via grouped sequential-to-parallel training." International Conference on Database Systems for Advanced Applications. Cham: Springer International Publishing, 2022.

[5] Zaccone, Riccardo, et al. "Speeding up heterogeneous federated learning with sequentially trained superclients." 2022 26th International Conference on Pattern Recognition (ICPR). IEEE, 2022.

[6] Lu, You-Ru, Xiaoqian Wang, and Dengfeng Sun. "Tackling Imbalanced Class in Federated Learning via Class Distribution Estimation." (2022).

[7] Chen, Dawei, et al. "Digital twin for federated analytics using a Bayesian approach." IEEE Internet of Things Journal 8.22 (2021): 16301-16312.

[8] Nguyen, John, et al. "Where to begin? Exploring the impact of pre-training and initialization in federated learning." NeurIPSW (2022).

[9] Chen, Hong-You, et al. "On the importance and applicability of pre-training for federated learning." The Eleventh International Conference on Learning Representations. 2022.

[10]  Li, Tian, et al. "Federated optimization in heterogeneous networks." Proceedings of Machine learning and systems 2 (2020): 429-450.

[11] Karimireddy, Sai Praneeth, et al. "Scaffold: Stochastic controlled averaging for federated learning." International conference on machine learning. PMLR, 2020.

[12] Durmus Alp Emre Acar, Yue Zhao, Ramon Matas Navarro, Matthew Mattina, Paul N Whatmough, and Venkatesh Saligrama. Federated learning based on dynamic regularization. International Conference on Learning Representations, 2021.

[13] Reddi, Sashank, et al. "Adaptive federated optimization." ICLR (2021).

[14] Li, Qinbin, Bingsheng He, and Dawn Song. "Model-contrastive federated learning." Proceedings of the IEEE/CVF conference on computer vision and pattern recognition. 2021.

[15] Karimireddy, Sai Praneeth, et al. "Mime: Mimicking centralized stochastic algorithms in federated learning." Advances in Neural Information Processing Systems, 2021 .

[16] Qu, Zhe, et al. "Generalized federated learning via sharpness aware minimization." International Conference on Machine Learning. PMLR, 2022.

[17] Caldarola, Debora, Barbara Caputo, and Marco Ciccone. "Improving generalization in federated learning by seeking flat minima." European Conference on Computer Vision. Cham: Springer Nature Switzerland, 2022.

[18] Wang, Jianyu, et al. "Slowmo: Improving communication-efficient distributed sgd with slow momentum." ICLR (2020).

[19] Kairouz, Peter, et al. "Advances and open problems in federated learning." Foundations and Trends® in Machine Learning 14.1–2 (2021): 1-210.

**Questions:**

**Questions**
1. Regarding the client selection strategy,
    * How does the model behave on out-of-distribution (OOD) clients, *i.e.*, the clients having maximum distance from the global distribution sketch?
    * How often are clients having larger distances from the global distribution sketch selected for training with respect to clients having more similar distributions?
    * Referring to 3) in Weaknesses, what happens when all local distributions are very distant from the global one, i.e. if $p_i - p_j \approx 0 \forall i,j\in[L]$? (referring to the notation in Algorithm 3)
    * How does this approach compare with other smart client selection strategies? Please see 1) in Weaknesses for examples on related works.
2. Regarding experiments in Figure 2:
    * The proposed method seems to be particularly effective on the Shakespeare dataset rather than on MNIST/FEMNIST. Is this behavior more related to the dataset  itself, or to the task? Is the one-pass distribution sketch more effective for NLP? Experiments on an additional vision or NLP dataset could help clarify this behavior. Examples are federated CIFAR10/100, or StackOverflow.
    * The text mentions the experiments were run for 200 rounds (line 315), but in the plots the X axes show a different number of rounds for each experiment. Which is the correct information?
3. How are the values of $B$ and $R$ (Alg. 1) selected? Are they hyperparameters? Which values were chosen? How sensitive is the method to such a choice?
4. Is there a risk of privacy leakage from the distribution sketch if no Laplace noise is added?

**Minor questions** (did not affect my rating)
1. How many clients were used in the experiments?
2. What is the average number of samples per client?

**Suggestions**
1. I would suggest revising the abstract (lines 10-14) to obtain a more effective result. For instance, it is not clear how the distribution sketch helps in the client selection, or how this method helps training on new unlabeled clients. A few additional details might send the message better.

**Minor suggestions** (did not affect my rating)
* Typos:
    * line 45: *is* computational ... --> "are"
    * In Sec. 3.2, Algorithm **3** from Sec. D appears referenced as Algorithm **0**.
    * lines 332-334 probably needs rephrasing (specifically referring to the sentence "Once we have the global model")
* Def 2.1 introduces $p_1$, $p_2$ and $D$, whose set of belonging is not formally defined - *e.g.*, $\mathbb{R}$.

**Limitations:**

Main limitations were addressed by the authors.

Additional limitation: potential lack of fairness towards users belonging to less represented distributions.

---

> ### Author Rebuttal · Authors · 2023-08-09
>
> We thank the reviewer for the comments and suggestions to improve our paper! Please see the following clarifications.
>
> **Reference**
>
> We thank the reviewer for a great summary of the literature. We have updated the paper by including the mentioned reference in the Introduction and Related Work section.
>
> **Client selection strategy**
>
> A major difference between our approach with other smart client selection strategies is that we do not need the loss/gradient of the candidate clients. In other words, we avoid the transmission of the global model to non-selected clients and loss/gradient computation on them.
>
>
> **Probability of selecting distant client**
>
> Consider the following example based on Figure 1. If we set the sketch of A_2 to be the global sketch. Then the distance of A_0, A_1, A_3, and A_4 to the global sketch would be [0.16,0.11,0.14,0.18]. Following Algorithm 3, we mark their selection probability as [0.04745458, 0.8129602 , 0.11588869, 0.02369653]. The distant client, A_3, will be sampled with probability 0.02369653.
>
>
> **All distant clients**
>
> As shown in Algorithm 3, if all clients are far away from the global sketch, then $p_i$ remains similar across clients. In this case, after the normalization step, the client selection process is equivalent to uniform sampling.
>
>
> **More Experiments**
>
> We conducted CIFAR-10 experiments using 100 clients, where L=3K and K=2. The client dataset follows a non-iid distribution, as specified in the split provided at https://github.com/yjlee22/FedShare. We visualized the iteration-wise convergence of FedAvg with and without the proposed client selection (please refer to the rebuttal PDF). The results demonstrate that our distribution sketch improves the test accuracy convergence rate, leading to faster iterations.
>
> **Rounds in experiments**
>
> We introduce early stopping to prevent overfitting. Feel free to refer to line 321 of the paper for the details.
>
> **Choice of Parameters**
>
> We suggest choosing $B$ and $R$ as follows: $R$ should be chosen proportional to $O(\sqrt{n})$, where $n$ is the maximum possible client data size. We set $B$ to be $O(\log(n))$ following theoretical guarantees in [1].  As a result, our sketch size should be significantly smaller than the dataset size.
>
>
> **Privacy leakage from the distribution sketch if no Laplace noise is added**
>
> Yes. To understand why, consider a client with an extreme outlier in its dataset. This outlier will likely be the only point in one of the RACE partitions, meaning that we could potentially localize it from the sketch alone, given access to the sketch's randomness. There are several ways to deal with this (e.g. using the distributed private sketch framework of [2]), but the addition of Laplace noise is the simplest method.
>
>
>
> **Experiments settings**
>
> For FEMNIST and shakespeare, we follow the exact settings in the LEAF https://leaf.cmu.edu/ project.  For MNIST, there are 30 clients in total. For the client selection, we set our parameters following Section 5.3 (line 315 to 320).
>
> We thank the reviewer for the comments on the writing. We have updated the paper accordingly for a better presentation.
>
> [1] Locality-Sensitive Hashing Scheme Based on p-Stable Distributions
>
> [2] Differentially-Private Multi-Party Sketching for Large-Scale Statistics

---

> > ### Comment · Reviewer_iXQ3 · 2023-08-14
> > **Rebuttal reply**
> >
> > I thank the authors for addressing my questions and running additional experiments on CIFAR10.
> >
> > - Regarding the comparison with other client selection strategies, I would like to point out that not all of them require the local loss or gradient to be communicated from the clients to the server. Please refer to [1] for a recent review of such strategies. Since the results presented in Fig. 2 show that the one-pass distribution sketch is not particularly effective on all datasets (eg., FEMNIST), I would like to understand how it compares with other existing selection strategies. For instance, how does clustered federated learning behave in the presented scenarios (Sec. 4.6.1 in [1])? And approaches based on maintaining fairness (Sec. 4.6.4)? And methods requiring the exchange of loss/gradient (eg., Power of choice [2])? Which are the pros and cons of these methods when compared with the one here proposed?
> >
> >
> > - As also confirmed by the authors, this approach is not effective in realistic cross-device scenarios, in which each client likely holds a local data distribution potentially very different from the global one, thus falling back into the uniform sampling. Could the authors comment on the amount of clients that this algorithm is actually able to handle? More precisely: how many clients can this approach handle before falling back into the uniform sampling? Is there a threshold? Which are the realistic scenarios in which the authors see this approach applicable? This limitation should be explicitly addressed in the paper.
> >
> >
> >
> > [1] Smestad, Carl, and Jingyue Li. "A Systematic Literature Review on Client Selection in Federated Learning." arXiv preprint arXiv:2306.04862 (2023).
> >
> > [2] Cho, Yae Jee, Jianyu Wang, and Gauri Joshi. "Client selection in federated learning: Convergence analysis and power-of-choice selection strategies." arXiv preprint arXiv:2010.01243 (2020).

---

> > > ### Author Response · Authors · 2023-08-16
> > > **We thank the reviewer for the comments. Please see the following clarifications.**
> > >
> > > We thank the reviewer for the comments. Please see the following clarifications.
> > >
> > > **Q1: Client Selection strategy**
> > >
> > > We updated the cifar10 experiments with a baseline using the strategy in [1] suggested by [2]. We observe that both our method and Fedavg reach the test accuracy 45.6, while [1] gives a 44.5 best test accuracy. Moreover, our method converges significantly faster than [1] and FedAvg.
> > >
> > > Next, we would like to provide an analysis that has been added to the revision of the paper.
> > >
> > > 1. Clustering approaches:  We have two primary concerns pertaining to the clustering approaches: (1) Computation: The existing clustering approaches necessitate $O(n^2)$ operations, with $n$ representing the number of available clients. In contrast, our approach involves only a linear traversal of all clients, resulting in superior efficiency in terms of overall amortized running time. (2) Assumptions: Another significant concern arises from the fact that, in our specific context, client selection occurs among active clients. During each round, these active clients constitute only a subset of the entire dataset and could potentially be randomly chosen. Consequently, the active clients might not encompass all clusters within each round. This limitation implies that performing sampling over each cluster might not be attainable within this particular setup.
> > >
> > >
> > >
> > > 2. Fairness in client selection: We do not prioritize fairness within the scope of this paper. Nonetheless, our distance measure could be potentially useful for fairness applications, as it enables the identification or sampling of atypical clients by identifying clients with the greatest distances within a global sketch. We would like to emphasize that our sketching algorithm is an efficient and private way to quantify distribution distance. We have demonstrated utility for cold-start and heterogeneous training, but there are likely further applications (which are beyond the scope).
> > >
> > > 3. Client selection required loss/weight/gradient: In our research, our primary emphasis lies in leveraging sketches for quantifying data heterogeneity.  For the non-selected clients, computation of loss/weight/gradient would introduce extra cost in federate optimization. Moreover, if required, the communication of weight/gradient will share the burden with the network, which is against one important motivation of client selection: to reduce communication load by selecting less useful clients for training. We would like to combine the data sketching with the local model status in the future work.
> > >
> > >
> > >
> > >
> > >
> > >
> > >
> > >
> > > **Q2: As also confirmed by the authors, ...**
> > >
> > >
> > > We think there may be a misunderstanding here. If all client distributions are **equally** far away from the global distribution, then all $p_i$ in Algorithm 3 will be the same and we will have uniform sampling. One way for this to happen is for each client distribution to be infinitely faraway from that of every other client. The other way is for each client to have exactly the same distribution. The first case is a degenerate situation where none of the clients share any part of their input domain. The second case is the IID setting with no heterogeneity.
> > >
> > > Neither situation happens in realistic cross-device scenarios - even though a client distribution may be far from the global distribution, some clients will still be closer / further than others. Because of the softmax normalization on line 17 of Algorithm 3, it is only the *relative distance* that matters. Our sampling distribution is non-uniform even when these distances are very large (though it asymptotically approaches the uniform distribution as the smallest pairwise distance approaches infinity).
> > >
> > > The equidistant case (where we default to uniform sampling) becomes increasingly unlikely as the number of clients increases. We do not observe this problem in our tests on Shapespeare (1129 clients) and FEMNIST (3550 clients). In fact, in practice, the number of clients scale to millions (https://ai.googleblog.com/2017/04/federated-learning-collaborative.html), so we do not expect this to pose problems in practice.
> > >
> > > We apologize for not making this point more clear in our original response.
> > >
> > > [1]Client Selection in Federated Learning under Imperfections in Environment
> > >
> > > [2]Smestad, Carl, and Jingyue Li. "A Systematic Literature Review on Client Selection in Federated Learning." arXiv preprint arXiv:2306.04862 (2023).’

---

> > > > ### Comment · Reviewer_iXQ3 · 2023-08-18
> > > > **Answer to authors**
> > > >
> > > > I thank the authors for their clarifications and the additional comparison.
> > > >
> > > > I believe the work to be valuable for the FL community and I'm happy to raise my score to 6.

---

### Official Review · Reviewer_QJNz · 2023-07-05

**Soundness:** 3 good
**Presentation:** 3 good
**Contribution:** 3 good
**Rating:** 7
**Confidence:** 4

**Summary:**

This paper proposed to use sketching for measuring heterogeneity in user data distributions in federated learning settings. The sketching is based on locality sensitive hashing and the authors have shown that in theory the measurement from the count sketch converges to the true kernel density estimation. The authors showed that the heterogeneity measurement and the count sketch can be used to 1) improve convergence by sampling users smartly according to their density in the distribution and 2) improve personalization by warming up the new client with a known nearest client’s model. The effectiveness of the approaches have been empirically tested on simple benchmark FL datasets.


**Strengths:**

* Measuring data heterogeneity has been an important problem to solve and the authors proposed a simple, intuitive yet effective idea. The implementation is easy to extend to existing FL frameworks.
* The experiments demonstrated not only the effectiveness of heterogeneity measurement but also how to apply the measurement for improving federated learning convergence and personalization, showing the general applicability of the proposed methods.


**Weaknesses:**

* For cross-device setting, some algorithms (e.g. with differential privacy) might incur large communication costs depending on the size of the sketch matrix.
* The datasets and models used in the experiments are very small scale and some do not reflect the true heterogeneity in the wild.
* Figure 1 can be explained better with the label distribution for users in group A. E.g. Do users with higher dissimilarity have dissimilar digits in MNIST and vice versa?
* Minor points about writing:
1) RACE introduced in section 2 has matrix $A \in {R}^{B\times R}$ while the sketch matrix in the following paragraph has transposed the axes. There would be less friction of reading if they are consistent.
2) Line 140, algorithm 0 should be algorithm 4.


**Questions:**

* What are the values for B and R chosen in this paper?
* How should one choose these values for different datasets?


**Limitations:**

There is no or minor potential negative societal impact of this paper. The authors have acknowledged limitations in the appendix of the paper.

---

> ### Author Rebuttal · Authors · 2023-08-09
>
> We thank the reviewer for the comments and suggestions to improve our paper! Please see the following clarifications.
>
> **Choices of parameters**
>
> We suggest choosing $B$ and $R$ as follows: $R$ should be chosen proportional to $O(\sqrt{n})$, where $n$ where
>  is the maximum possible client data size. We set $B$ to be $O(\log(n))$ following theoretical guarantees in [1].  As a result, our sketch size should be significantly smaller than the dataset size.
>
>
> **Figure 1 can be explained better with the label distribution for users in group A. E.g. Do users with higher dissimilarity have dissimilar digits in MNIST and vice versa?**
>
> Thank you for the suggestion. Datasets in group A are drawn randomly under a Dirichlet distribution, and datasets in group B are drawn randomly under a uniform distribution. As a result, the label distribution among the datasets in Group A is more heterogeneous than that of Group B. As shown in the histogram, the pairwise distances among group A are smaller because they include all the digits. On the other hand, the pairwise distances among group B are larger and more inconsistent because they are likely to include different digits.
>
> **CIFAR-10 results**
>
> We conducted CIFAR-10 experiments using 100 clients, where L=3K and K =2. The client dataset follows a non-iid distribution, as specified in the split provided at https://github.com/yjlee22/FedShare. We visualized the iteration-wise convergence of FedAvg with and without the proposed client selection. The results demonstrate that our distribution sketch significantly improves the test accuracy convergence rate, leading to faster iterations.
>
>
> [1] Locality-Sensitive Hashing Scheme Based on p-Stable Distributions
>
> We thank the reviewer for the comments on the writing. We have updated the paper accordingly for a better presentation.

---

### Official Review · Reviewer_Jfur · 2023-07-06

**Soundness:** 2 fair
**Presentation:** 3 good
**Contribution:** 1 poor
**Rating:** 3
**Confidence:** 3

**Summary:**

The authors propose a private strategy for measuring data heterogeneity using a RACE matrix. The RACE matrix essentially measures the normalized frequency of data points being assigned to particular LSH codes. These LSH count matrices are sent up to a server where it is applied into two different regimes:\\
1. For distribution-aware client selection, the server samples clients with probability directly associated to closeness between the local RACE and global RACE matrix.
2. For cold-start learning, a new client receives a personalized client model whose distribution matrix is closest to the new client's distribution matrix.

The approach is assessed over federated training on several datasets. Local accuracies are compared by using the global or locally closest model across both warm and cold clients.

**Strengths:**

-The RACE matrix is theoretically a good estimator of DTV between distributions.
-Scheme for developing client selection probabilities is simple.
-There are apparently not many existing algorithms for addressing cold start learning.
-Tested over multiple datasets with promising local accuracies.

**Weaknesses:**

1. The RACE matrix is apparently already well-known as an excellent kernel density estimator. The contribution seems to be application of it to the federated setting, which has limited novelty.
2. Under the suggested client scheme, the global model is favoring the largest set of homogeneous clients (popularity bias). Clients with atypical data distributions will have a low chance of being selected for global averaging, so there is no benefit for them to participate. If a client with an atypical distribution were to be randomly selected, their model would be washed out with a global model which is, on average, favoring distributions unlike theirs.
3. The RACE estimation could be heavily affected by quantity skew (which will accompany label-distribution skew in practice). Many entries will be under-filled if a local dataset simply isn't large enough and the output space of the hash function larger than 2.
4. In the cold start setting, the central server needs to be storing every possible client model. This would not be scalable. Such approach might better fit decentralized learning.
5. MNIST and FEMNIST are too easy to classify. CIFAR-10 training, at the minimum would have been nice to see, which suffers far more from label-distribution heterogeneity, see: "Federated Learning on Non-IID Data Silos: An Experimental Study," Li et al. (2021).

**Questions:**

1. I can't seem to easily find the settings for the LSH setup. How many LSH functions were used per sketch? Were they refreshed after every averaging round? Does the server distribute the hash functions?
2. Were there any measurement regarding how the global model accuracy is affected by personalized model selection in the cold client setting? I imagine there should be a classical tradeoff between global and personalized accuracy.



**Limitations:**

No limitations discussed. Please consider including drawbacks.

---

> ### Author Rebuttal · Authors · 2023-08-09
>
> We are glad that you find the method simple and promising, and we have done our best to address your questions carefully. We hope that you might consider raising the score in light of our response.
>
> **Q1: Our contribution with respect to the RACE matrix**
>
> The RACE matrix, indeed, proves to be an exceptional KDE technique. However, its application has been restricted to single probability distributions, both in theory and practice. This research aims to extend the capabilities of the RACE matrix by leveraging it as a distribution sketch, enabling the assessment of divergence across multiple distributions. It is important to emphasize that efficient divergence estimation is a nontrivial task. For instance, most sampling-based density estimators would require high-dimensional integrals involving all pairwise interactions between samples from the two clients. As a result, there exists a notable distinction between the conventional use of RACE for kernel density queries, as seen in the existing literature, and its novel application for approximating statistical distances, which is one of our key contributions.
>
> **Q2: Clients with atypical data distributions will have a low chance of being selected for global averaging, so there is no benefit for them to participate. If a client with an atypical distribution were to be randomly selected, their model would be washed out with a global model which is, on average, favoring distributions unlike theirs.**
>
> Atypical distribution clients generally destabilize the training process due to data heterogeneity [1,2]. We agree that sampling clients with distributions close to the goal distribution might enhance the popularity bias and is related to the important topic of fairness in federated learning. However, our distance measure can be used in several different ways, one of which could be to identify/sample atypical clients by looking at the largest distance with a global sketch. We want to emphasize that while we focus on personalized FL, our sketching algorithm provides an efficient and effective way to measure distribution differences in a private way, which is helpful for many different goals: stabilize training, fairness, personalized FL, etc.
>
>
> **Q3: Effect of quantity skew**
>
> The RACE approximation of the density is an unbiased estimator whose variance (error) decreases with the client data size. RACE can be thought of as a quantization (histogram) of the data distribution, where the quantization’s granularity reduces and accuracy improves as more data arrive. While the accuracy will be better for high-quantity clients than low-quantity ones, we do not expect the under-filling of the matrix to pose serious problems because we are not computing density point estimates. We use RACE to approximate the statistical distance (a robust global quantity that depends on the full support of the distribution) rather than the point density (a more sensitive local quantity that depends on the presence/absence of nearby data). The overall RACE distance will still be meaningful (and unbiased), even if the underlying density estimates are only coarsely quantized.
>
>
>
> **Q4: In the cold start setting, the central server needs to store every possible client model**
>
> In our approach, the server only stores the distribution sketch of existing clients, which is small in memory. The cold-start client will find its nearest neighbor in terms of distribution sketch in the server. Next, the cold-start client can retrieve the personalized model from its nearest neighbor client. As a result, the server doesn’t have to maintain all the client models.
>
>
>
> **Q5: CIFAR-10 results**
>
> We conducted CIFAR-10 experiments using 100 clients, where L=3K and K=2. The client dataset follows a non-iid distribution, as specified in the split provided at https://github.com/yjlee22/FedShare. We visualized the iteration-wise convergence of FedAvg with and without the proposed client selection (see the submitted PDF). The results demonstrate that our distribution sketch significantly improves the test accuracy convergence rate, leading to faster iterations.
>
>
> **Q6: LSH setup**
>
> We suggest using $R=O(\sqrt{n})$ LSH functions globally for sketching, where $n$ is the maximum possible client data size. We set $B$ to be $O(\log(n))$ following theoretical guarantees in [3].  We assume the server distributes the same set of hash functions to all clients by sharing the random seed. We don’t need to refresh the hash functions. In our experiment, we use sign random projections (Image) and MinHash (Text) as LSH functions. The server does not need to distribute the hashes, as their parameters can be generated from a single random seed.
>
> **Q7: Global model accuracy in the cold client setting.**
>
> In this setting, we have trained a global model based on a set of clients. Each client participating in the training has labeled data. Next, for every trained client, we fine-tune the global model on its own data and get a personalized model associated with this client. Given a cold-start client with no label information on its client data, we generate its data distribution sketch and identify the trained client with the closest sketch distance. Finally, the cold-start client retrieves the model from the selected trained client as the personalized cold-start model. We note that the global model on the server is not affected by this procedure.
>
> [1] Li, Tian, et al. "Federated optimization in heterogeneous networks." Proceedings of Machine learning and systems 2 (2020): 429-450.
>
> [2] Gao, D., Yao, X., & Yang, Q. (2022). A survey on heterogeneous federated learning. arXiv preprint arXiv:2210.04505.
>
> [3] Locality-Sensitive Hashing Scheme Based on p-Stable Distributions

---

> > ### Comment · Reviewer_Jfur · 2023-08-19
> > **Response to authors**
> >
> > I thank the reviewers for their detailed response, especially the CIFAR-10 experiments
> >
> > Interestingly, reviewer iXQ3 and I share similar concerns and questions, but I am divergent in my overall opinion of the work based on responses.
> >
> > 1. Popularity bias concerns
> > >We agree that sampling clients with distributions close to the goal distribution might enhance the popularity bias and is related to the important topic of fairness in federated learning.
> >
> > Reducing bias against atypical clients is an inherent component of the learning process and accommodating such client data as opposed to washing it out will result in a more robust global model. In fact, FL is uniformly trending towards using heterogeneous data to the global and local model's benefit. So a question to the authors: why would atypical clients participate in a training process when their distribution is far away from an averaged global distribution?
> >
> > Indeed, using the RACE as an identifier of atypical global distributions would be an interesting topic of study but that is not the primary focus of this paper and a separate algorithm would need to be developed to handle atypical clients.
> >
> > 2. Cold start storage
> >
> > >Finally, the server retrieves the personalized model on the nearest neighbor client and sends it to the new client.
> >
> >  If the server is not storing client models, then it is effectively querying the nearest neighbor for its model – can this scale to a large grid of impersistent clients (the usual assumption), since you are now possibly performing **two rounds** of communications with a given client?
> > (Acquiring its local model to build the global model, and another round to ask for it again if it needs to go to a neighbor.) Clients are randomly sampled (similar to the setup in the paper) to simulate clients dropping on and off a grid. Note that two rounds of communication are unusual for a centralized FL communication protocol and results in **increased communication complexity**.
> >
> > 3. LSH functions
> > >We suggest using $R=\mathcal{O}(\sqrt{n})$ LSH functions globally for sketching, where $n$ is the maximum possible client data size. We set to be $\mathcal{O}({\log(n)})$ following theoretical guarantees in [3].
> >
> > What are the $B$, $R$ values and data splits used for the experiments? (What $n$ are you typically encountering per your Dirichlet allocation?)
> >
> > At this time, I am maintaining my score.

---

> > > ### Author Response · Authors · 2023-08-21
> > > **Thank you for your comments. Please see the following clarification.**
> > >
> > > Dear Reviewer  Jfur
> > >
> > > Thank you for your comments. We would like to clarify that the primary focus of our paper is to develop a primitive sketching algorithm to efficiently and privately measure the divergence between high-dimensional distributions. We emphasize that an efficient method to compute such a “dataset distance” is extremely valuable to various applications. In the paper, we demonstrate this utility on two applications within Federated Learning. RACE can also be used to directly estimate the TVD, as suggested by Reviewer c2sU, or to identify clients with atypical distributions (by considering clients to be atypical if they have a higher-than-typical TVD / sketch distance to the global distribution). This does not require a change to our distance estimation algorithm but is rather an additional way to apply the techniques we introduce in the paper.
> > >
> > > 1. **Washout atypical clients.**
> > >
> > > We think there may be a misunderstanding here. Our algorithm does not encourage bias toward atypical clients. We reweigh clients that are closer to the global distribution (and are thus selected with higher sampling probability), in order to maintain an unbiased objective function.
> > >
> > > To investigate the role of atypical clients, we conducted a new experiment on Cifar10 where we “wash out” the 1% - 10% clients whose RACE distances are furthest from the total client distribution. If we randomly sample from the remaining active clients in each round, we see similar slower convergence and performance as FedAvg. This corresponds to the situation where we identify atypical clients as those with distances smaller than the 1% or 10% percentile thresholds on the distance histogram and exclude them from training. This result suggests that our algorithm’s sampling strategy performs better than thresholding. We would like to point out that it is difficult to explicitly decide whether a client is “atypical” in terms of absolute distance to the global distribution. Moreover, an important sampling framework with an unbiased objective function benefits theoretical analysis and practical deployment.
> > >
> > > 2. **Cold start storage**
> > >
> > > We argue that in cold start settings, one round of communication only requires transmitting the sketch instead of the model. Typical sketch sizes are an order of magnitude smaller than the model, so this does not significantly increase the communication cost. For instance, in the Shakespeare model, the race sketch is 10kB, while the LSTM model is 1.07MB. Moreover, we can perform additional floating-point / integer encoding and quantization tricks, which can achieve a 2-3x compression over the 10kB sketch size. The cold start client $A$ first sends its sketch to the server. Then the server uses $A$’s sketch as a query to its nearest neighbor sketch, which represents client $B$. Finally, client $B$ sends its local model to client $A$. We note that transmitting the sketch is much lighter than transmitting the model.
> > >
> > >
> > >
> > > 3. **LSH Parameters.**
> > >
> > >  We can give a concrete example here. In Shakespeare dataset, we expect the $n$ to be more than $10^4$ (mean samples per client: 3,743.2, std: 6,212.26). We set $R$ to be $100$ and $B$ to be $32$ or $64$. Our RACE sketch is experiencing good scalability with large client data sizes and high dimensional and sparse data such as text.
> > >
> > > We value your conversation about the sketching and sampling aspects. We kindly request your consideration of adjusting the score in light of our response.

---

### Author Rebuttal · Authors · 2023-08-09

We thank reviewers [R1(Jfur), R2(QJNz), R3(iXQ3), R4(c2Su)] for their thoughtful and highly supportive feedback! We were glad that the reviewers found the problem significant and interesting [R2 R3], the observations and theoretical analysis insightful and valuable [R3, R4], the methods novel and clear [R2, R3, R4], and the experimental results convincing and promising [R2 R3].

We have updated the paper to incorporate constructive suggestions. We summarize the attached additional experiments:

[ R1,  R2, R3] Addition CIFAR-10 experiments on client sampling.

[R4] Simulation on TVD estimation.

---

### Decision · Program_Chairs · 2023-09-21

**Decision:**

Accept (poster)

**Comment:**

This paper an interesting method to measure heterogeneity across client data in federated learning. The reviews are generally positive and agree that this will add value to the FL literature.